# New and Effective Inhibitor of Class I HDACs, Eimbinostat, Reduces the Growth of Hematologic Cancer Cells and Triggers Apoptosis

**DOI:** 10.3390/pharmaceutics17040416

**Published:** 2025-03-25

**Authors:** Pavel Spirin, Valeria Vedernikova, Tatsiana Volkava, Alexey Morozov, Alla Kleymenova, Anastasia Zemskaya, Lena Shyrokova, Yuri Porozov, Ksenia Glumakova, Timofey Lebedev, Maxim Kozlov, Vladimir Prassolov

**Affiliations:** 1Department of Cancer Cell Biology, Engelhardt Institute of Molecular Biology, Russian Academy of Sciences, Vavilova 32, 119991 Moscow, Russia; vedernikova.vo@phystech.edu (V.V.); kglumakova@mail.ru (K.G.); lebedevtd@gmail.com (T.L.); 2Center for Precision Genome Editing and Genetic Technologies for Biomedicine, Engelhardt Institute of Molecular Biology, Russian Academy of Sciences, Vavilova 32, 119991 Moscow, Russia; 3Moscow Center for Advanced Studies, Kulakova 20, 123592 Moscow, Russia; 4Faculty of Biology, Ludwig Maximilians University, Großhaderner Str. 2, 82152 Munich, Germany; t.volkava@campus.lmu.de; 5Laboratory of Regulation of Intracellular Proteolysis, Engelhardt Institute of Molecular Biology, Russian Academy of Sciences, Vavilova 32, 119991 Moscow, Russia; runkel@inbox.ru; 6Laboratory of Molecular Basis of Action of Physiologically Active Compounds, Engelhardt Institute of Molecular Biology, Russian Academy of Sciences, Vavilova 32, 119991 Moscow, Russia; kaamail@rambler.ru (A.K.); a.zemskaia@mail.ru (A.Z.); kozlovmavi@gmail.com (M.K.); 7Department of Experimental Medical Science, Lund University, 221 84 Lund, Sweden; olena.shyrokova@med.lu.se; 8St. Petersburg School of Physics, Mathematics, and Computer Science, HSE University, 199106 Saint Petersburg, Russia; yuri.porozov@gmail.com; 9Advitam Laboratory, Mihaila Shushkaloviħa 13, 11030 Belgrade, Serbia

**Keywords:** epigenetic, HDACs, belinostat, lymphoma, leukemia, apoptosis, cell cycle

## Abstract

**Background:** Histone deacetylases (HDACs) are critical epigenetic modulators involved in regulating various molecular mechanisms essential for cell development and growth. Alterations in HDAC activity have been linked to the progression of numerous cancers, including lymphoma. Over the past decade, the FDA has approved several HDAC inhibitors for lymphoma treatment, leading to heightened interest in this emerging class of drugs. **Methods:** In our research, we developed a novel HDAC inhibitor that exhibits high selectivity for class I HDACs. **Results:** Our in vitro findings indicate that treating lymphoma/leukemia cells with this inhibitor results in a marked suppression of cell growth and promotes apoptosis, while leaving the cell cycle unaffected. **Conclusions:** We propose that our new inhibitor, named eimbinostat, holds significant promise as a potential therapeutic agent for the treatment of hematologic malignancies such as lymphoma or leukemia.

## 1. Introduction

Hematologic cancers refer to malignancies that affect the blood, bone marrow, and lymphatic system. These cancers can affect the body in various ways, leading to symptoms such as fatigue, infections, anemia, and bleeding issues. Treatment options often include chemotherapy, radiation therapy, targeted therapy, and stem cell transplants [1,2,3].

Lymphoma, a type of blood cancer, primarily develops within the lymphatic system. Treatment options vary based on the specific subtype of lymphoma and typically involve a combination of radiotherapy and chemotherapy [4,5]. While Hodgkin’s lymphoma (HL) and non-Hodgkin’s lymphoma (NHL) treatments have shown relatively high efficacy, the risk of relapse remains significant. This underscores the urgent need for the development of novel and effective therapeutic agents for lymphoma.

Zinc-dependent HDACs are crucial epigenetic regulators classified into four distinct classes based on their function and cellular localization. Class I HDACs (HDAC1–3) are mainly found in the nucleus, whereas HDAC8 (also a class I enzyme) and regulatory HDACs 4, 5, 7, and 9 from class IIa can move between the nucleus and the cytoplasm. Class IIb HDACs (HDAC6 and HDAC10) and class IV HDACs (HDAC11) primarily operate in the cytoplasm [6]. In addition to their involvement in histone modification, HDACs play a role in regulating gene expression as part of transcription elongation complexes [7]. Abnormal activity of HDACs can promote tumor progression and contribute to resistance against treatment by influencing oncogene expression [8]. Alterations in HDAC function or expression have been observed across various cancer types, often correlating with advanced disease stages and poor prognosis [9]. Class I nuclear HDACs play a significant role in leukemia by functioning as components of repressor complexes such as CoREST, Sin3, MiDAC, and NuRD, all of which include HDAC1/2. Notably, SMRT/NCoR is the only complex that operates with HDAC3 [10,11]. It is well established that CoREST is involved in leukemogenesis [12,13,14], and elevated levels of Sin3 have been associated with poor prognosis in acute myeloid leukemia [15]. Both MiDAC, which is crucial for chromosome alignment during mitosis [16], and NuRD, whose inhibition leads to growth arrest and apoptosis [17], are recognized as promising anti-cancer targets. Additionally, the pro-survival activity of SMRT/NCoR is linked to its roles in cell cycle progression, genomic stability, and apoptosis/survival pathways in leukemia-derived cell lines [18].

The role of HDACs in hematological malignancies’ development is gaining significant attention, as these epigenetic modulators present promising targets for the treatment of lymphoma [19]. Consequently, HDAC inhibitors (HDACi) have emerged as attractive agents for altering gene expression patterns in cancer cells. Several HDAC inhibitors have received FDA approval for treating different lymphoma subtypes [20]. For instance, the pan-HDAC inhibitor belinostat has been approved for patients with relapsed or refractory peripheral T-cell NHL [21]. Nevertheless, the non-selective characteristics of pan-HDAC inhibitors frequently result in significant toxicity and serious side effects, underscoring the urgent demand for new, selective HDAC inhibitors [22,23].

Investigating the specific roles of individual HDACs in cancer progression is essential; therefore, development of novel selective HDAC inhibitors is of significant interest. HDAC inhibitors, including belinostat, romidepsin, panobinostat, entinostat, and vorinostat, have emerged as important therapeutic agents in the treatment of leukemia and lymphoma by targeting various classes of HDACs. Belinostat primarily inhibits class I and class II HDACs, notably HDAC1, HDAC2, and HDAC6 [24,25]. Romidepsin is a class I-selective inhibitor that focuses on HDAC1 and HDAC2, making it particularly effective for cutaneous T-cell lymphoma [26]. Panobinostat is a pan-HDAC inhibitor that targets multiple classes, including class I (HDAC1, HDAC2, HDAC3) and class II (HDAC6), and is used in treating multiple myeloma [27]. Entinostat selectively inhibits class I and II HDACs, especially HDAC1 and HDAC2, with potential applications across various cancers [28]. Vorinostat, another broad-spectrum inhibitor affecting class I and II HDACs, is approved for cutaneous T-cell lymphoma. Other notable inhibitors like trichostatin A and sodium butyrate also target multiple HDAC classes. These inhibitors can influence gene expression by enhancing histone acetylation, which leads to the reactivation of silenced tumor suppressor genes and the induction of apoptosis in cancerous cells. This presents a promising approach for the treatment of leukemia and lymphoma. As research continues to explore their mechanisms and optimal combinations with other therapies, HDAC inhibitors hold significant promise for improving outcomes in these hematologic malignancies.

Selective pharmacological inhibition or targeted degradation/knockout of class I HDACs has been shown to induce apoptosis and cell cycle arrest in cancer cells [29,30,31]. Recently, N’-propylhydrazide analogs of hydroxamic pan-HDAC inhibitors like panobinostat and belinostat have been identified as potent and selective class I HDAC inhibitors with unique anti-cancer properties [23,32,33,34].

In this study, we synthesized and studied a novel N’-butylhydrazide analog of belinostat, termed eimbinostat, and compared its activity with hydrazostat, N’-propylhydrazide analog of belinostat, and belinostat by itself. Our in vitro findings demonstrate that treatment with eimbinostat significantly suppresses lymphoma cell survival through the induction of apoptosis.

## 2. Materials and Methods

### 2.1. Chemistry

#### 2.1.1. Reagents and Technical Equipment

Nuclear magnetic resonance (NMR) spectra (d, ppm; J, Hz) were registered on an AMX III-400 spectrometer (Bruker, Billerica, MA, USA) at 400 MHz for 1H NMR (internal standard: Me4Si; solvent: DMSO d6) and 100.6 MHz for 13C NMR with suppression of carbon-proton interaction (solvent: DMSO d6). High-resolution mass spectra (HRMS) were registered on a Bruker Daltonics micrOTOF-Q II hybrid quadrupole time-of-flight mass spectrometer using electrospray ionization (ESI); measurements were performed in positive ion mode. Thin layer chromatography (TLC) was performed on the plates of Kieselgel 60 F254 (Merck, Darmstadt, Germany) and the column chromatography was carried out on silica gel Kieselgel, 0.035–0.070 mm, (Acros Organics, Geel, Belgium) using the elution systems mentioned in the text.

#### 2.1.2. General Procedure for the Preparation of Eimbinostat

(i)Schiff base preparation—Belinostat hydrazide (HB) (900 mg, 2.84 mmol), prepared as described earlier [33], was suspended in *i*-PrOH (20 mL), then butyraldehyde (410 mg, 5.68 mmol) and AcOH in a catalytic amount (20 mg) were added, and the mixture was stirred for 18 h and used as-is in the next step.(ii)Eimbinostat—NaBH_4_ (500 mg, 13.2 mmol) was added to obtain the above suspension and the mixture was boiled for 1.25 h. After cooling to an ambient temperature, the reaction mixture was diluted with water (20 mL), neutralized with AcOH to pH~6.0, and evaporated to the final volume 15 mL. The supernatant was decanted, the gummy residue was solidified by twice evaporation with H_2_O (from 10 to 20 mL), with subsequent cooling in ice for 30 min and rubbing. The crude product was purified by silica gel chromatography using a mixture of two eluents: (a) n-hexane–EtOAc (1:2) and (b) CHCl_3_-EtOH (10:1) in equal proportions, affording eimbinostat in the yield of 756 mg (71.4%). 1H NMR (300 MHz, DMSO) δ 10.31 (s, 1H), 9.65 (s, 1H), 7.92 (s, 1H), 7.77 (d, J = 7.7 Hz, 1H), 7.71 (d, J = 8.1 Hz, 1H), 7.58 (t, J = 7.8 Hz, 1H), 7.45 (d, J = 15.9 Hz, 1H), 7.24 (t, J = 7.8 Hz, 2H), 7.10 (d, J = 7.5 Hz, 2H), 7.03 (t, J = 7.3 Hz, 1H), 6.59 (d, J = 15.8 Hz, 1H), 5.06 (s, 1H), 2.73 (t, J = 6.8 Hz, 2H), 1.36 (qt, J = 13.8, 6.7 Hz, 5H), 0.89 (t, J = 7.1 Hz, 3H). 13C NMR (75 MHz, DMSO) δ 163.65 (s), 140.76 (s), 137.95 (s), 137.15 (s), 136.39 (s), 132.41 (s), 130.47 (s), 129.65 (s), 127.52 (s), 125.26 (s), 124.77 (s), 122.95 (s), 120.81 (s), 51.31 (s), 30.17 (s), 20.19 (s), 14.33 (s). ESI-HRMS (*m*/*z*): Calcd for C_19_H_23_N_3_O_3_S [M + H]^+^ 374.1533, found 374.1514

### 2.2. Bioinformatics Analysis of Gene Expression and Gene Fitness Data

Belinostat AUC values were taken from the CTRPv2 drug screen [35] and data were downloaded from DepMap database. Gene expression data for 19,145 tumor samples were collected from 144 datasets downloaded from R2: Genomics Analysis and Visualization Platform (http://r2.amc.nl, accessed on 21 February 2025) and grouped by cancer type using a previously described algorithm [36,37]. Gene expression data from all tumor samples were utilized to compute z-scores for each sample corresponding to each HDAC gene, followed by the calculation of mean z-scores for both ALL and lymphoma. RNAi and CRISPR screening data regarding gene fitness were obtained from the DepMap database and processed as previously described [37]. In brief, dependency scores for each cell line were derived using the DepMap database version 22Q2, averaging gene scores from RNAi (Achilles + DRIVE + Marcotte, DEMETER2) [38,39] and CRISPR (DepMap 22Q2 Public + Score, Chronos) [40] screens, leading to the calculation of mean scores for each cancer type. Using these values, z-scores were computed for each tumor type to assess how the sensitivity of each cell type compared to others. Mean gene expression and gene fitness data were clustered using the Ward D2 algorithm. Statistical analyses and data processing were conducted in Python 3.11.5, while heatmaps and data clustering were generated in RStudio 4.2.3 with the ComplexHeatmap package 2.22.0 [41].

### 2.3. Biology

#### 2.3.1. Cell Cultures

The following cell lines were used: Jurkat (human T-lymphoblastic leukemia), Ramos (human B-cell lymphoma/Burkitt’s lymphoma), Mv4;11 (acute myeloid leukemia), Karpas 299 (human T-cell lymphoma), and Hut-78 (cutaneous T-cell lymphoma). Cells were cultured in RPMI-1640 medium supplemented with either 10% or 20% fetal bovine serum (FBS). The growth medium was supplemented with 1 mM sodium pyruvate, 2 mM L-glutamine, and antibiotics, including streptomycin (100 μg/mL) and penicillin (100 units/mL), sourced from Thermo Fisher Scientific, Waltham, MA, USA. Cultures were kept at 37 °C in a humidified environment with 5% CO_2_. Mv4;11 and Jurkat cells were gifted by the Heinrich–Pette Institute—Leibniz Institute for Experimental Virology. Ramos (Acc No. 85030802), Karpas 299 (Acc No. 06072604), and Hut-78 (Acc No. 88041901) cells were purchased from Merck, Darmstadt, Germany. The identity of all cell lines was verified using the Short Tandem Repeat (STR) DNA Genotype Analysis.

#### 2.3.2. Cell Viability Analysis

To assess cell viability, Mv4;11 cells were plated at a density of 2500 cells/well in a total volume of 200 μL per well in a 48-well plate. Jurkat, Ramos, and Karpas 299 cells were plated at densities of 5000 cells/well in the same volume, while Hut-78 cells were plated at a density of 10,000 cells/well in a total volume of 200 μL per well. Cells were exposed to belinostat, hydrazostat, or eimbinostat at different concentrations between 0 and 12 μM, using a volume of 50 μL for each treatment. All drugs were dissolved in DMSO, so a corresponding concentration of DMSO (0.024%) was included as a control to match the maximum drug concentration of 12 μM. After incubating for 72 h, cell viability was assessed using a manual count with a Neubauer chamber after staining with 0.4% solution of Trypan blue (Invitrogen Corp., Carlsbad, CA, USA). The half-maximal inhibitory concentrations (IC50s) were calculated using nonlinear regression analysis with variable slope fitting, performed with GraphPad Prism software version 8.4.3 (GraphPad Software, San Diego, CA, USA).

#### 2.3.3. Gene Expression Analysis

Total RNA was extracted using a Trizol reagent (Invitrogen, Carlsbad, CA, USA). The synthesis of complementary DNA was carried out with the RevertAid First Strand cDNA Synthesis Kit (Thermo Fisher Scientific, Waltham, MA, USA). Quantitative polymerase chain reaction (qPCR) was conducted in triplicate utilizing SYBR Green (Evrogen, Moscow, Russia) and analyzed with the CFX96 Touch Real-Time PCR Detection System (Bio-Rad, Hercules, CA, USA). The primer sequences are provided in Appendix A. Cycle threshold values were normalized against the endogenous control gene glyceraldehyde-3-phosphate dehydrogenase (GAPDH).

#### 2.3.4. Western Blot Analysis and Acetylation Efficiency Assay

Western blot analysis was conducted following a previously published protocol] [42]. For the α-tubulin acetylation assay, cells were lysed using a lysis reagent (Promega, Madison, WI, USA). For the histone H3 acetylation assay, cells were lysed with TEB buffer (0.5% Triton X-100, 2 mM PMSF, and Protein Safe Protease Inhibitor Cocktail from TransGen, Beijing, China). The resulting lysates were centrifuged, and the pellet was washed and resuspended in 0.2 M HCl, then incubated overnight at 4 °C. Following centrifugation, the supernatant was neutralized with 1.5 M Tris-HCl (pH 8.8). Proteins from the supernatant were separated by electrophoresis on polyacrylamide gels (PAAG) of appropriate percentages and subsequently transferred onto a nitrocellulose membrane. The membrane was blocked with 5% dry milk (Bio-Rad, Hercules, CA, USA) in PBST for 60 min at room temperature. Primary antibodies against acetylated K9/14 histone H3 (1:2000), total histone H3 (1:4000), K40 acetylated α-tubulin (1:3000), and total α-tubulin (1:7000) were applied to the membrane and incubated overnight at 4 °C, followed by washing with PBST. Next, a horseradish peroxidase-conjugated secondary antibody specific to the primary antibodies (1:8000) was added to the membrane and incubated for 60 min at room temperature. After additional washing with PBST, the signal was visualized using an Immobilon Western Kit (EMD Millipore Corporation, Burlington, MA, USA) and captured using a ChemiDoc Imaging System (Bio-Rad, Hercules, CA, USA).

The following antibodies were utilized in the study: rabbit anti-histone H3 (9715S) and rabbit anti-acetylated K9/14 histone H3 (9677S) from CST (Danvers, MA, USA); rabbit anti-α-tubulin K40 acetylated (sab5600134) from Abcam (Waltham, MA, USA); and mouse anti-α-tubulin (T5168) from Sigma (Livonia, MI, USA). Horseradish peroxidase-conjugated secondary antibodies, specifically anti-mouse (sc-2005) and anti-rabbit (sc-2004), were sourced from SCBT (Dallas, TX, USA).

Primary antibody against HDACs: anti-HDAC1 rabbit Ab 2062S (Cell Signaling Technology, Danvers, MA, USA); anti-HDAC2 rabbit Ab 2540S (Cell Signaling, USA); anti-HDAC3 mouse mAb 3949S (Cell Signaling, USA); anti-HDAC8 rabbit mAb 66042S (Cell Signaling, USA). The secondary HRP-labeled anti-rabbit (Abcam, Cambridge, UK) or anti-mouse conjugates antibodies (Enzo, Farmingdale, NY, USA) were used for HDAC detection.

#### 2.3.5. HDAC Selectivity Assay Using the Cell-Test System (s3CTS)

The cell-test system (s3CTS) based on the application of HCT116 cells was used to study the selectivity of HDAC inhibitors [43]. Cells were plated in 96-well culture plates at a density of 1.5 × 10^4^ cells per well. After 24 h, when the cells reached 90–100% confluence, they were treated with various concentrations of the test inhibitors for an additional 24 h. Following this incubation, three quarters of the medium from each well was removed and replaced with an equivalent volume of fresh medium containing the same concentration of the inhibitor along with one of three substrates (Sub^Ac/Pro/Tfa^) at a concentration of 30 μM. After an additional 4 h incubation, aliquots of the culture medium were transferred to a black plate designed for fluorescence measurements (SPL Life Sciences, Pocheon, Republic of Korea). These samples were double-diluted with a solution of 2 mg/mL trypsin in Tris-HCl buffer (pH 8) and incubated for 60 min at 37 °C. Fluorescence was then measured using a Spark multifunctional plate reader (Tecan Trading, Mannedorf, Switzerland) at excitation/emission wavelengths of 360/470 nm. The fluorescence intensity for each well was normalized based on the cytotoxicity data obtained from the same well. The average normalized fluorescence for each concentration of the test compounds was calculated using the formula provided below. RFU = Σn [(Fi − F0)/Cv]/n; where Fi is the fluorescence value in a test well, F0 is the fluorescence value in a well with medium and without cells, Cv is the cell viability, and n is the number of replicates.

#### 2.3.6. Analysis of Apoptosis and Cell Cycle

To assess the percentage of apoptotic cells and cell cycle disruption, Jurkat, Hut-78, and Ramos cells were plated as previously described and treated with belinostat, hydrazostat, and eimbinostat at IC50 and IC75 concentrations. After 72 h of incubation, cells were double-stained with Annexin V-FITC (Invitrogen, Thermo Scientific, Waltham, MA, USA) and propidium iodide (PI) (Sigma-Aldrich, St. Louis, MO, USA). Alternatively, cells were fixed in ethanol and stained with PI for apoptosis or cell cycle distribution analysis. All measurements were performed on an LSR Fortessa flow cytometer (BD Biosciences, San Jose, CA, USA). Apoptosis rates were analyzed using FlowJo software version 10.0.7 (FlowJo LLC, Ashland, OR, USA), while cell cycle distribution was assessed with ModFit LT software version 4.1.7 (Verity Software House, Topsham, ME, USA).

#### 2.3.7. Molecular Modeling and Docking

The X-ray model of HDAC3, PDB ID 4A69, was selected to perform molecular docking and binding modeling. Preliminary preparation of the model was performed, and B, C, and D chains, crystal water, acetic acid, and ligands were removed. The A chain of the protein and the Zn ion present in the active center were taken for molecular docking. The X-ray structure of HDAC6 of Danio rerio (PDB ID 5EEN) [44,45] was chosen as a reference model of belinostat binding [44,45]. Target 4A69 was processed in the Protein Preparation Wizard module, which completed the missing side chains of some amino acids, added hydrogen, and performed an initial structure minimization using the OPLS4 force field [46]. The structures of belinostat and eimbinostat were constructed for molecular docking. Each molecule was validated and optimized in the LigPrep procedure. Molecular docking of the prepared ligands into the active site of HDAC3, PDB ID 4A69, was performed. In the Receptor Grid Generation procedure, the center of the region was set near Zn. The ion itself was used to set the metal coordination constraint and the Hbond/metal constraint, since from the work of E. Bülbül et al. [47] it is known that inhibitors of this protein chelate zinc in the active site upon binding. We applied the procedure of flexible induced fit molecular docking [48] taking into account possible ligand conformers, local mobility of the main chain, side chains of the active center amino acids, and zinc.

#### 2.3.8. Data and Statistical Analysis

All experiments were performed in triplicate. Statistical significance was determined with *p* < 0.05. Statistical analyses were conducted using GraphPad Prism software version 8.4.3 (GraphPad Software, San Diego, CA, USA). The types of used statistical tests are indicated in the figure legends.

## 3. Results

### 3.1. Synthesis and Characterization of Eimbinostat

Reductive alkylation of belinostat hydrazide (HB) was used for the synthesis of eimbinostat as published earlier for N′-propylhydrazide analog preparation [33] (Figure 1). The use of isopropanol at both stages of synthesis simplified its implementation and dramatically increased the yields of the intermediate Schiff base and the final product. As a consequence, eimbinostat isolated by silica gel chromatography was of enough purity to be used in biological experiments without additional purification. The structure of eimbinostat has been unequivocally confirmed by NMR spectra and high-resolution mass spectra (HRMS) (Appendix A).

### 3.2. Bioinformatics Assessment of the Effects of HDAC on Cell Viability

Drug sensitivity data from the CTRPv2 study [35] show that blood cancers, especially of lymphoid origin, display selective sensitivity to HDAC inhibitor belinostat (Figure 1A). Pan-cancer analysis of HDAC gene expression in tumors of different origins from R2: Genomics Analysis and Visualization Platform showed mostly uniform expression of all HDAC genes. HDAC1, 2, 3, and 6 were the most ubiquitously expressed, while other HDAC genes had a lower expression for most tumor types. However, none of the genes had distinct tissue-specific expression (Figure 1B). Gene fitness analysis based on DepMap data, which shows how cell survival and proliferation are affected by gene knockdown/knockout revealed that depletion of most HDAC genes does not significantly affect cell proliferation, except HDAC3, which seems essential to all cell types (Figure 1C). However, ALL and lymphoma cells showed more pronounced sensitivity to HDAC1 depletion (Figure 1D). Across all HDAC genes which are expressed at higher than medium levels in ALL and lymphoma, HDAC1 has the most impact on cell proliferation (Figure 1E).

### 3.3. Molecular Docking

Molecular docking calculations of belinostat with HDAC3, PDB ID 4A69, showed that this molecule is able to form a coordination bond with Zn in the active center of the protein. Moreover, the ligand stacking was exactly the same as in the 5EEN reference model, which indicates the adequacy and applicability of the computational models and methods we used. The docking score for this binding method was −9.117 kkal/mol. In addition, hydrogen bonds are formed with HIS134, GLY296, and GLY143, as well as pi-pi interactions with PHE144, PHE200, and HIS172. Belinostat forms a chelate bond between the carbonyl group and Zn. The binding is also due to a significant contribution of hydrophobic interactions (Figure 2A,C). Flexible docking of eimbinostat with the HDAC3 target, PDB ID 4A69, showed a docking score of −9.62 kkal/mol. Ligand stacking was essentially the same as that of the 5EEN reference. The ligand formed hydrogen bonds with ASP93, GLY 143, and HIS134 targets. Pi-pi interactions were formed with HIS22 and PHE200. Carbonyl oxygen was involved in the coordination of Zn (Figure 2B,D).

### 3.4. Eimbinostat Significantly Suppresses Growth of Lymphoma Cells and Selectively Inhibits Class I HDAC Deacetylation Activity

In this study, we compared the effects of eimbinostat with the well-known FDA-approved pan-HDAC inhibitor belinostat and the recently developed class I HDAC inhibitor hydrazostat [23] (Figure 3A).

Initially, we assessed the relative expression of HDACs in leukemia and lymphoma cell lines. Our analysis revealed that the expression levels of HDAC class I genes (HDAC1, HDAC2, HDAC3, and HDAC8) in Jurkat, Hut-78, and Mv4;11 cells were significantly higher compared to the lymphoma cell lines Ramos and KARPAS 299 (Figure 3B). The relative protein levels of HDACs in studied cells were also detected (Figure 3C). The lowest level of HDAC2 and HDAC3 was found in Ramos cells and the lowest content of HDAC1 and HDAC2 were detected in Hut 78 and KARPAS 299 cells.

Next, we evaluated the cytotoxicity of belinostat, hydrazostat, and eimbinostat, finding that cells with relatively elevated HDAC class I expression were more sensitive to eimbinostat, a selective inhibitor of HDAC class I (Appendix A). Notably, all studied cell lines demonstrated high sensitivity to the cytotoxic effects of belinostat.

For our experiments, we selected three cell lines with varying levels of HDAC class I expression: Jurkat (high), Ramos (low), and Hut-78 (medium). Interestingly, Jurkat cells exhibited the highest sensitivity to the cytotoxic effects of eimbinostat (Figure 4A).

We compared the inhibitory effects of the HDAC inhibitors belinostat and eimbinostat at concentrations corresponding to IC_50_ and 2× IC_50_. The effectiveness of HDAC inhibition was assessed by measuring the accumulation of acetylated forms of two protein substrates: α-tubulin (α-TubK40ac) and histone H3 (H3K9/14ac). It is well established that deacetylation of α-TubK40ac is primarily mediated by HDAC6 (class IIb), while H3K9/14ac is provided mainly by HDAC class I enzymes. Treatment with the pan-HDAC inhibitor belinostat resulted in a significant upregulation of both α-tubulin and histone H3 acetylation. In contrast, eimbinostat selectively inhibited HDAC class I activity without significantly affecting α-TubK40ac levels (Figure 4B), and enhancing only H3K9/14ac accumulation (Figure 4C).

In addition to Western blot analysis, the selectivity of eimbinostat for class I HDACs has been validated using the cell-test system (s3CTS), which employs three subtype-selective fluorogenic substrates with the general structure Boc-Lys(Acyl)-AMC, where acyl = propionyl (Sub^Pro^, HDACs class I), acetyl (Sub^Ac^, HDACs class I and IIb) and trifluoroacetyl (Sub^Tfa^, HDACs class IIa) added to HCT116 cells [43]. We demonstrated that the deacylation of Sub^Pro^ was nearly fully inhibited by eimbinostat and hydrazostat. However, both compounds did not entirely eliminate the deacetylation of Sub^Ac^, as some activity from HDAC6 class IIb remained, and neither eimbinostat nor hydrazostat had any impact on the signal level for Sub^Tfa^ (Figure 4D). Such a pattern of Sub^Pro/Ac/Tfa^ deacylation inhibition curves is characteristic of HDAC1/2/3 selective inhibitors, for example, UF010 and CI-994. The results of belinostat testing confirmed its multipotent inhibitory activity with a preference for action against HDACs class I/IIb (Figure 4D), as previously shown [43].

### 3.5. Eimbinostat Induces Apoptosis

Subsequently, we investigated the induction of apoptosis following treatment with HDAC inhibitors across all studied cell lines (Figure 5A–C). For those cells, the cells treated with inhibitors were stained with Annexin V-FITC and PI followed by FACS analysis. Cells stained with Annexin V and not stained with PI were interpreted as cells in early apoptosis (EA). Cells stained both with Annexin V and PI (A+/PI+) were interpreted as cells in late apoptosis (LA). PI-positive cells and not stained with Annexin V (A−/PI+) were interpreted as necrotic cells (N). Jurkat and Hut-78 cells exhibited high sensitivity to apoptosis induced by both the pan-HDAC inhibitor belinostat and the HDAC class I inhibitors hydrazostat and eimbinostat (Figure 5A,B). In contrast, Ramos cells showed reduced sensitivity to apoptosis induction caused by HDAC class I inhibitors compared to Jurkat and Hut-78 cells (Figure 5C). Only relatively high concentrations of these inhibitors, specifically at IC_75_, induced apoptosis. Treatment of Ramos cells with belinostat significantly activated apoptotic pathways.

Interestingly, Jurkat and Hut-78 cells with relatively higher levels of HDAC class I expression compared to Ramos are more sensitive to apoptosis induction. This suggests that the more efficient apoptosis induction caused by eimbinostat and hydrazostat may be linked to their selective inhibition of HDAC class I enzymes (HDAC1, 2, 3, and 8) involved in survival.

### 3.6. Eimbinostat Does Not Affect Cell Cycle

Next, we examined the effects of HDAC inhibitors on the cell cycle. Our analysis revealed that eimbinostat did not impact cell cycle transitions (Figure 6). In contrast, significant alterations in cell cycle distribution were observed following treatment with belinostat (Figure 6A–C). Notably, these alterations varied among the studied cell lines. In Jurkat cells treated with belinostat, we observed a decrease in the percentage of cells in the G1 phase and an increase in the S phase (Figure 6A,B), which may indicate a suppression of the G1-to-S phase transition. Conversely, treatment of Hut-78 cells with belinostat resulted in an increased percentage of cells in the G1 phase, suggesting suppression of the G1 phase transition (Figure 6C). Interestingly, while hydrazostat did not affect the cell cycle in Jurkat or Hut-78 cells, it caused a significant accumulation of cells in the G1 phase when applied to Ramos cells (Figure 6D). These findings indicate that the range of potential targets affected by eimbinostat is distinct not only from those impacted by the pan-HDAC inhibitor belinostat but also from those influenced by the selective HDAC class I inhibitor hydrazostat. Furthermore, the effects of eimbinostat may vary significantly depending on the type of cells to which the inhibitors are applied.

### 3.7. Treatment with Eimbinostat Alters Expression of Genes Associated with Leukemia/Lymphoma

To identify potential target genes affected by the HDAC inhibitors, we analyzed the expression of several genes involved in the regulation of apoptosis and the cell cycle. mRNA was isolated from cells after 72 h of incubation with inhibitors at IC50 concentrations and analyzed using qRT-PCR. While no consistent patterns in gene expression changes were observed across all treated cells, significant suppression of anti-apoptotic genes *Bcl-XL* (*BCL2L1*) and *Bcl-2* was detected in Hut-78 cells following treatment (Figure 7). Furthermore, we examined the expression of genes encoding Cyclins A1, B1, D2, and the key cell cycle regulator E2F1, noting a significant upregulation of *CCNA1* encoding Cyclin A1 in Jurkat and Hut-78 cells treated with eimbinostat. In summary, the compounds studied primarily exhibit similar effects on gene expression across various cell lines, with the exception of a few specific targets, such as *Bcl-2* and *CCNA1*.

## 4. Discussion

Pan-HDAC inhibitors have been shown to suppress the survival of cancer cells from various origins [49,50]. Numerous pan-HDAC inhibitors are currently under investigation in preclinical and clinical trials. However, the specific roles of individual HDACs in diverse biological processes are often difficult to interpret [51] in part because of significant variability in the effects of HDAC inhibitors across different cell types. Most mechanisms underlying this variability are not yet fully understood, highlighting the need for effective methods to selectively target HDACs. The development of chemical agents that specifically target certain HDACs or classes of HDACs is of considerable interest, both for research purposes and as promising therapeutic agents. Recently, two N’-propylhydrazide analogs of hydroxamic pan-HDAC inhibitors, panobinostat and belinostat, were identified as potent and selective HDAC class I inhibitors with unique anti-cancer properties [23,32].

In this study, we evaluated a newly synthesized N’-butylhydrazide analog of belinostat—eimbinostat—which significantly reduced cell survival. Our findings indicate that treatment with this HDAC inhibitor induces apoptosis of leukemia/lymphoma cell lines in vitro. Notably, the novel HDAC class I selective inhibitors eimbinostat and hydrazostat demonstrated a more pronounced induction of apoptosis in cells with relatively high expression levels of *HDAC1/2/3* compared to the pan-HDAC inhibitor belinostat. Importantly, treatment with eimbinostat led to alterations in the expression of genes encoding proteins responsible for cell survival. Specifically, we observed a significant decrease in the expression of anti-apoptotic genes *Bcl-XL* and *Bcl-2* in lymphoma cells treated with eimbinostat, which may partially account for the observed induction of apoptosis. Notably, cell apoptosis is associated with cell cycle arrest. In our study, the cell cycle profile was not affected by eimbinostat. To distinguish these two processes and to explore the effects of eimbinostat on the cell cycle in non-apoptotic cells we performed a cell cycle analysis solely on viable cells, intentionally excluding the cells in the sub-G1 population. The total culture analysis indicated a significant increase in the percentage of apoptotic cells, suggesting that cell cycle progression in living cells remains unaffected.

According to existing literature, the N’-propyl substituent in benzohydrazide-derived HDAC inhibitors provides strong inhibition of the HDAC3 isoform [52]. In contrast, the presence of a butyl group results in less potent inhibition but offers more balanced inhibition between HDAC1 and HDAC3 [34,52]. This difference in properties among homologous inhibitors likely stems from structural features within the foot pocket of the enzyme’s active site that accommodates the alkyl residue of the hydrazide’s zinc-binding group (ZBG) [53]. Comparing the mechanisms by which belinostat and its hydrazide analogs exert anti-cancer effects allows for an assessment of the individual roles of HDAC1 and HDAC3 in the survival of specific leukemia and lymphoma cell types. Given that hydrazostat exhibited more pronounced suppressive activity against Hut-78 and Ramos cells compared to eimbinostat, it is reasonable to infer that HDAC3 is a key molecular target for inducing cell death. This works in good correspondence with the bioinformatics analysis suggesting the significant role of HDAC3 in the survival of leukemia/lymphoma cells (Figure 1E).

The structural similarities between these inhibitors suggest comparable efficiency in cellular uptake and similar biotransformation pathways. Thus, the potency of HDAC3 inhibition emerges as a critical factor in suppressing cell viability. The greater flexibility of the embinistat molecule allowed the realization of a significantly more advantageous ligand arrangement in the active center of the target, which is confirmed by the better docking score. At the same time, the extended structure of eimbinostat explains the increased selectivity revealed in our experiments compared to belinostat. These structural features open a great field for further development of compounds of this class. This hypothesis aligns well with recent findings highlighting the importance of effectively targeting HDAC3 to overcome resistance to pan-HDAC inhibitors in diffuse large B-cell lymphoma [54].

In conclusion, we propose that eimbinostat holds significant promise for both basic research and potential clinical applications. The most important limitation of our study is that we have only recently initiated investigations into the effects of eimbinostat in in vivo models. Future research will include comprehensive in vivo studies, an in-depth exploration of potential off-target effects, and an analysis of the molecular mechanisms that this drug may modulate. These efforts will enable to provide a more precise assessment of the eimbinostat potential for clinical application in treating malignant hematopoietic diseases.

## Data Availability

The data presented in this study are available on request from the corresponding author.

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
