# Peer review of "New and Effective Inhibitor of Class I HDACs, Eimbinostat, Reduces the Growth of Hematologic Cancer Cells and Triggers Apoptosis"

_pharmaceutics, 2025, doi:10.3390/pharmaceutics17040416_

Round 1
Reviewer 1 Report
Comments and Suggestions for Authors
x/c/
The manuscript by Spirin et al. focuses on a new class I HDAC inhibitor called Eimbinostat and its effects on hematologic cancer cells. This is a novel N’-butylhydrazide analog of Belinostat. The research shows that Eimbinostat reduces the growth of lymphoma and leukemia cells and induces apoptosis, while not affecting the cell cycle. The study compares Eimbinostat to Belinostat (a pan-HDAC inhibitor) and Hydrazostat (another class I HDAC inhibitor). Key findings show that Eimbinostat selectively inhibits class I HDACs, suppresses lymphoma cell survival and induces apoptosis, and does not significantly alter the cell cycle. Eimbinostat is shown to alter the expression of genes associated with leukemia/lymphoma.
The main impact is that this new epigenetic drug candidate is shown to have promise as a potential therapeutic agent for hematologic malignancies. This clearly written study presents a comprehensive set of data that includes details on the synthesis and characterization of Eimbinostat, bioinformatics analysis of HDAC gene expression, molecular docking studies, and in vitro experiments on various lymphoma and leukemia cell lines. In the Introduction, there is no need to explain the nature of the various cancer types discussed, there is a typo in line 45 in the word cancers. There are sentences on lymphoma that sound repetitive. Details about the mechanisms of action (l.113-115) can be deleted or moved to the Discussion. The methods need an explanation as to how the identity of the commercial cell lines used was validated. Figures 1A-1D need to be presented in high resolution and with a larger font size to increase readability.
The authors mention that the non-selective nature of pan-HDAC inhibitors often leads to high toxicity and severe side effects, highlighting the critical need for novel, selective HDAC inhibitors. However, although this is an in vitro study, the authors should have a paragraph in the discussion discussing potential toxicities of Eimbinostat and future in vitro and in vivo studies as part of future directions in the drug development pathway.
Author Response
The manuscript by Spirin et al. focuses on a new class I HDAC inhibitor called Eimbinostat and its effects on hematologic cancer cells. This is a novel N’-butylhydrazide analog of Belinostat. The research shows that Eimbinostat reduces the growth of lymphoma and leukemia cells and induces apoptosis, while not affecting the cell cycle. The study compares Eimbinostat to Belinostat (a pan-HDAC inhibitor) and Hydrazostat (another class I HDAC inhibitor). Key findings show that Eimbinostat selectively inhibits class I HDACs, suppresses lymphoma cell survival and induces apoptosis, and does not significantly alter the cell cycle. Eimbinostat is shown to alter the expression of genes associated with leukemia/lymphoma.
The main impact is that this new epigenetic drug candidate is shown to have promise as a potential therapeutic agent for hematologic malignancies. This clearly written study presents a comprehensive set of data that includes details on the synthesis and characterization of Eimbinostat, bioinformatics analysis of HDAC gene expression, molecular docking studies, and in vitro experiments on various lymphoma and leukemia cell lines.
Answer
We are grateful to the review for the constructive analysis of our manuscript and have introduced all necessary corrections and clarifications. We are sure that this has really improved our manuscript.
Point 1
In the Introduction, there is no need to explain the nature of the various cancer types discussed, there is a typo in line 45 in the word cancers.
Answer
Thank you, we agree and excluded this description from our manuscript.
Point 2
There are sentences on lymphoma that sound repetitive. Details about the mechanisms of action (l.113-115) can be deleted or moved to the Discussion.
Answer
We agree, checked the text for repetitions and deleted the section mentioned by the Reviewer.
Point 3
The methods need an explanation as to how the identity of the commercial cell lines used was validated.
Answer
Thank you for this important remark, the information was added to the methods section.
Point 4
Figures 1A-1D need to be presented in high resolution and with a larger font size to increase readability.
Answer
We checked the resolution of this Figure and additionally uploaded this Figure in high resolution in Zip archive using MDPI submission system.
Point 5
The authors mention that the non-selective nature of pan-HDAC inhibitors often leads to high toxicity and severe side effects, highlighting the critical need for novel, selective HDAC inhibitors. However, although this is an in vitro study, the authors should have a paragraph in the discussion discussing potential toxicities of Eimbinostat and future in vitro and in vivo studies as part of future directions in the drug development pathway.
Answer
Thank you very much. The paragraph describing the limitations of our work, including the section describing the need of animal models that will be used in future experiments was added to conclusion part of our manuscript.
Reviewer 2 Report
Comments and Suggestions for Authors
pharmaceutics-3518047
Histone deacetylase (HDAC) inhibitors have been popularly used for clinical/preclinical research in a wide range of cancers. In which, there are some pan-HDAC inhibitors have obtained FDA approval for treatment of hematologic malignancies (leukemia and lymphoma), including belinostat, romidepsin, panobinostat, etinostat, and vorinostat. Nevertheless, the non-selective target of pan-HDAC inhibitors results in high toxicity and severe side effects, requiring the exploration of novel and selective HDAC inhibitors. In this study, Spirin et al. developed a HDAC inhibitor (eimbinostat), which exerted higher selective potency than previous pan-HDAC inhibitors in multiple hematologic malignant cell lines. The authors also described the sensitivity of the malignant cells to eimbinostat mediated by the induction of apoptosis, while the cell cycle profile remained unaffected. Although the study elicited a novel, selective and effective therapy for treatment of hematologic malignancies, the authors should address all my concerns below prior to getting the acceptance of publication for the manuscript.
Major concerns
- The research did not have in vivo studies using eimbinostat to treat animals engrafted with malignant cells, as well as the toxicity testing in healthy animals. Although the authors mentioned this issue in the conclusion (line 542-548), I am not sure the lack of in vivo and toxicity testing can remain the manuscript as a good candidate for the journal. Therefore, I will leave the decision of this issue for the Editor.
- As the study did not have in vivo testing, I suggest the authors obtain primary (e.g. bone marrow cells) from transgenic mice developing one type of hematologic malignancy, and challenge the primary cells by eimbinostat, in comparison to other pan-HDAC inhibitors. The current cell models in this study only include cell lines, which seems to be not enough without in vivo studies.
- In 2.3.5, why did you use only HTC116 cell lines? It is not a hematologic malignant cell line.
- In 2.3.7, why was only HDAC3 used for molecular modeling and docking, rather than HDAC1 and HDAC2?
- In all Western blot panels, the authors must include graphs of blot quantity, together with p-value calculations.
- In Fig 4B, the western blot should include the data of hydrazostat, to compare with eimbinostat.
- In Figure 4D, which cells were used for that experiment?
- In Figure 5, the pink bar should be EA (early apoptosis).
- Important! As the western blot in Figure 4B,C showed both HDAC inhibitors exerted their potencies after 24 hours (as written in line 415), the authors must have apoptosis assay (Figure 5), cell cycle assay (Figure 6), and gene expression profile (Figure 7) in 24 hours, and 48 hours of the inhibitors treatment, to have more convincing conclusion about the effect of eimbinostat on cell apoptosis, cell cycle and gene expression.
- In Figure 7, how could you have the conclusion without negative control as DMSO? Moreover, it would be much better if the authors could have the western blot of BCL-XL, BCL-2, and cyclin A1 in the cells after the treatment with all three inhibitors.
- Paragraph line 536-540, the authors must discuss the data of HDAC3 molecular modeling and docking.
- Cell apoptosis is strongly associated with cell cycle arrest. However, in this study, the cell cycle profile was not affected by eimbinostat. The authors must discuss this phenomenon, providing relevant information supporting their observation.
Minor issue
- Line 45, “ancers” should be “cancers”
- Some sentences were missing citation.
Line 43-51, the whole paragraph did not have any citations.
Line 66, after “expression”, it must have citation(s).
Line 86, missing citations in the last sentence of the paragraph.
Line 87-107, there were not any citations or references or ID of FDA approvals.
Line 523, missing citation(s) after “HDAC3 isoform”.
- Line 89 do not need to repeat the full writing of Histone deacetylase; abbreviation should be enough.
- Line 124, what is NMR. Line 129, what is TLC?
- Is the method in 2.1.2 (ii) based on previous studies?
- In figure legends, the letter A, B, C, D … must be written in Bold style.
- Line 354, “Figure 2” must be written in Bold style.
- Line 378, it should be “Fig.S4A,B”.
Author Response
Reviewer 2
Histone deacetylase (HDAC) inhibitors have been popularly used for clinical/preclinical research in a wide range of cancers. In which, there are some pan-HDAC inhibitors have obtained FDA approval for treatment of hematologic malignancies (leukemia and lymphoma), including belinostat, romidepsin, panobinostat, entinostat, and vorinostat. Nevertheless, the non-selective target of pan-HDAC inhibitors results in high toxicity and severe side effects, requiring the exploration of novel and selective HDAC inhibitors. In this study, Spirin et al. developed a HDAC inhibitor (eimbinostat), which exerted higher selective potency than previous pan-HDAC inhibitors in multiple hematologic malignant cell lines. The authors also described the sensitivity of the malignant cells to eimbinostat mediated by the induction of apoptosis, while the cell cycle profile remained unaffected. Although the study elicited a novel, selective and effective therapy for treatment of hematologic malignancies, the authors should address all my concerns below prior to getting the acceptance of publication for the manuscript.
Answer
We are very grateful to the reviewer for a meaningful review of our work and a significant number of important comments that are of great importance. We introduced all the necessary corrections in according to reviewer comments into our manuscript and are sure that it became much better.
Major concerns
Point 1
The research did not have in vivo studies using eimbinostat to treat animals engrafted with malignant cells, as well as the toxicity testing in healthy animals. Although the authors mentioned this issue in the conclusion (line 542-548), I am not sure the lack of in vivo and toxicity testing can remain the manuscript as a good candidate for the journal. Therefore, I will leave the decision of this issue for the Editor. As the study did not have in vivo testing, I suggest the authors obtain primary (e.g. bone marrow cells) from transgenic mice developing one type of hematologic malignancy, and challenge the primary cells by eimbinostat, in comparison to other pan-HDAC inhibitors. The current cell models in this study only include cell lines, which seems to be not enough without in vivo studies.
Answer
We appreciate the reviewer’s comments regarding the absence of experiments involving laboratory animals. We acknowledge that without such studies, it is challenging to predict the therapeutic effects of the tested drug.
In our research, we did not use primary malignant cells from the bone marrow of transgenic mice due to their unavailability. Furthermore, our initial focus was on the effects of HDAC inhibitors specifically on lymphoma cells, which are primarily associated with the lymphatic system rather than the bone marrow. But we agree with the critical role of animal research in drug development. In response, we have initiated toxicological studies in laboratory animals and are actively working to establish xenograft lymphoma tumors in immunodeficient mice.
We also noted the absence of animal studies as a significant limitation in the concluding section of our article.
However, the main point of our research is the synthesis and investigation of a novel HDAC inhibitor's selectivity. We conducted a detailed study of its biological activity across a range of lymphoma and leukemia cell lines and we believe it holds considerable interest for the readers of Pharmaceutics, despite the lack of animal models.
The animal researches will be incorporated into our future studies. Our next publication will delve deeper into the biological effects of Eimbinostat, exploring the mechanisms of cell death induction and its potential as a therapeutic agent. Thank you for your understanding and support as we continue this important line of research.
Point 2
In 2.3.5, why did you use only HTC116 cell lines? It is not a hematologic malignant cell line.
Answer
We apologize for any confusion caused by the incomplete name of the method, which may have led to misunderstandings regarding its appropriateness. We have made the necessary corrections to both the title of the section and its content. In section 2.3.5, "Materials and Methods," we provide a detailed description of the HDAC Selectivity Assay utilizing the cell-test system (s3CTS). This is a previously published cellular reporter system designed to evaluate the selectivity of HDAC inhibitors, constructed based on HCT-116 cells.
It is important to note that such systems are commonly employed in similar studies, including various commercial systems based on different cell lines. While the biological effects of the same inhibitors can vary across different cell lines, the mechanistic impact on the target protein remains consistent, irrespective of the cell line's characteristics. This principle also underpins the use of cell-free systems for assessing the selectivity of HDAC inhibitors.
Additionally, we want to emphasize that this system serves as a complementary method for determining the selectivity of HDAC inhibitors, alongside selectivity analyses and acetylation assessments (Western blot assay) performed on hematopoietic cell lines, specifically Jurkat, Hut-78, and Ramos. The results obtained from selectivity determinations in these cell types align well with those derived from the 3CST system based on the use of HCT-116 cells, further indicating that the effectiveness of the studied inhibitors on HDAC is independent of the specific nature of the cells used.
Point 3
In 2.3.7, why was only HDAC3 used for molecular modeling and docking, rather than HDAC1 and HDAC2?
Answer
In section 3.2, titled "Bioinformatics Assessment of the Effects of HDAC on Cell Viability," we represent the results from our bioinformatics analysis where we identified HDAC3, a member of the class I HDAC family, as a key potential target. This comprehensive analysis utilize the data of impact of CRISPR\CAS9 and RNAi mediated knockdown of targeting genes on cell survival. We found that the inhibition of HDAC3 activity may be linked to the survival of malignant hematopoietic cells. Consequently, HDAC3 was selected for the docking assay.
Furthermore, data on the effect of alkyl substituent length in hydrazide inhibitors are contradictory and have not been systematically investigated. However, it is known that the active sites of Class I HDAC are almost identical. The main exception is that Ser113/Ser118 of HDAC1/2 is altered to tyrosine in HDAC3, which leads to a steric hindrance so that bulky functional-groups of inhibitors are inaccessible to the foot pocket [Yuxiang Luo et al, Int. J. Mol. Sci. (2020); http://dx.doi.org/10.3390/ijms21228828].Thus, the possibility of placing an Eimbinostat molecule in the active site of HDAC3 even more implies the same possibility for closely related HDAC1/2. In addition, HDAC3 was selected for docking first of all due to the proposed importance of its inhibition for the anti-leukemia activity of the Eimbinostat.
Point 4
In all Western blot panels, the authors must include graphs of blot quantity, together with p-value calculations.
Answer
We performed the needed calculations for the acetylation efficiency study (Western blots Figure 4). The ratios of acetylated to nonacetylated forms, normalized to control (DMSO), are represented as heatmaps. F.c. ‒ fold change. P-values were determined for each samples relative to control (K-) using Unpaired t-test, with asterisks indicating significance levels: ns (p > 0.05), ** (p < 0.01), *** (p < 0.001), **** (p < 0.0001).
We are agree with the reviewer and in this section, this significantly increase the perception of the results.
The Western blots on represented on Figure 3 C were obtained to confirm the presence of not only HDAC mRNA, but also protein. The results obtained did not imply any quantitative interpretation, so we decided to refrain from such a calculation.
Point 5
In Fig 4B, the western blot should include the data of hydrazostat, to compare with eimbinostat.
Answer
In our previous study, we described the synthesis of the drug hydrazostat and its selectivity profile (Vagapova et al, Biomedicines, 2021). Our studies revealed that hydrazostat exhibits a significant inhibitory effect on class I histone deacetylases. In the current work, we utilized hydrazostat as an additional reference to compare the cytotoxic efficacy of Eimbinostat, with other inhibitors, including belinostat and hydrazostat itself. It is important to note that this study does not aim to investigate the impact of hydrazostat on acetylation efficacy of hematopoietic cancer cell lines. However, to confirm its inhibitory properties, we present the results of its evaluation using a reporter cell system (s3CTS) (Figure 4D).
Point 6
In Figure 4D, which cells were used for that experiment?
Answer
In this section of our study, we employed a previously established system (3CTS) for evaluating the selectivity of HDAC inhibitors, which is based on HCT116 cells. This method served as an additional approach to validate the selectivity of the inhibitors we tested. To enhance clarity, we have revised Section 2.3.5 in the Materials and Methods to provide a more straightforward explanation of our methodology.
Point 7
In Figure 5, the pink bar should be EA (early apoptosis).
Answer
We are agree, corrected.
Point 8
Important! As the western blot in Figure 4B,C showed both HDAC inhibitors exerted their potencies after 24 hours (as written in line 415), the authors must have apoptosis assay (Figure 5), cell cycle assay (Figure 6), and gene expression profile (Figure 7) in 24 hours, and 48 hours of the inhibitors treatment, to have more convincing conclusion about the effect of eimbinostat on cell apoptosis, cell cycle and gene expression.
Answer
We sincerely appreciate the reviewer for highlighting this important point. We acknowledge that there was a typing error in the figure caption, which incorrectly stated that Western blots were performed after 24 hours of incubation. In fact, all experiments, including those related to western blot, cell cycle and apoptosis, were conducted after 72 hours of drug treatment. We have corrected this mistake in the caption and have also clarified the incubation times in other relevant sections of the text to ensure consistency and accuracy. Thank you for your understanding.
Point 9
In Figure 7, how could you have the conclusion without negative control as DMSO? Moreover, it would be much better if the authors could have the western blot of BCL-XL, BCL-2, and cyclin A1 in the cells after the treatment with all three inhibitors.
Answer
Thank you for your insightful comment. I would like to clarify that the results of the gene expression analysis conducted using the qRT-PCR method are presented in the form of a heatmap, normalized to cells treated solely with DMSO. This information is explicitly detailed in the figure caption. To enhance clarity, we have also made corresponding revisions in the main text of the manuscript.
Unfortunately, we did not observed common patterns of gene expression of studied genes, so we decided not to speculate on potential molecular mechanisms that might be altered in cells treated with drugs. In light of this, we think that protein analysis, which could confirm the observed changes, is not necessary at this stage of our research. Nonetheless, we felt it was important to present the obtained results to demonstrate that the drugs influence gene expression related to cell survival.
For a more comprehensive investigation of the underlying mechanisms, we plan to conduct a transcriptomic analysis of the treated cells. This approach will enable us to identify specific signaling pathways that may be affected by the tested inhibitors. Following that the protein analysis would indeed be relevant. Such work is intended for future phases of our study and was not the primary focus of this research.
Point 10
Paragraph line 536-540, the authors must discuss the data of HDAC3 molecular modeling and docking.
We are agree and included the paragraph addressing this comment.
Point 11
Cell apoptosis is strongly associated with cell cycle arrest. However, in this study, the cell cycle profile was not affected by eimbinostat. The authors must discuss this phenomenon, providing relevant information supporting their observation.
Answer
We agree with the reviewer that these two processes are indeed related. In our study, we designed the experiment to differentiate between these two processes and to investigate whether the Eimbinostat affect cell cycle in non-apoptotic cells. To achieve this, we conducted a cell cycle analysis exclusively on viable cells, excluding the subG1 population (cell cycle distribution was normalized on number of viable cells). Total culture analysis revealed a significant increase in the percentage of apoptotic cells. This finding suggests that the cell cycle progression in alive cells is not affected.
It is also important to note that cell cycle disturbances do not always lead to apoptosis. For instance, during G1 arrest induced by mitogenic factors, cells may repair rather than undergo apoptosis (Bruna Pucci et al., Neoplasia, 2000). Additionally, research has shown that c-Abl is essential for DNA damage-induced activation of MKK6 and p38, and that while MKK6 activation by c-Abl is necessary for c-Abl-induced apoptosis, it is not required for c-Abl-induced cell cycle arrest (Feng Cong et al., PNAS, 1999).
Minor issue
Point 1
Line 45, “ancers” should be “cancers”
Answer
Thank you for this comment. We corrected this misprint.
Point 2
Some sentences were missing citation.
Line 43-51, the whole paragraph did not have any citations.
Line 66, after “expression”, it must have citation(s).
Line 86, missing citations in the last sentence of the paragraph.
Line 87-107, there were not any citations or references or ID of FDA approvals.
Line 523, missing citation(s) after “HDAC3 isoform”.
Answer
We are grateful to the reviewer for this point. We added missing citations.
Point 4
Line 89 do not need to repeat the full writing of Histone deacetylase; abbreviation should be enough.
Answer
We agree and changed the full writing to abbreviation.
Point 5
Line 124, what is NMR. Line 129, what is TLC?
Answer
Thank you for your comment. We added this information to the manuscript.
Point 6
Is the method in 2.1.2 (ii) based on previous studies?
Answer
The synthesis of Eimbinostat was carried out according to the modified procedure described earlier for the synthesis of Hydrazostat [Kozlov M, et al Bioorganic Med. Chem. Lett. 2019], in particular, transformations (i) and (ii) were carried out in isopropanol without changing the solvent.
Point 7
In figure legends, the letter A, B, C, D … must be written in Bold style.
Answer
We are grateful to the reviewer for this point. We checked the manuscript and introduced the necessary corrections.
Point 8
Line 354, “Figure 2” must be written in Bold style.
Answer
Thank you for your comment. We introduced the necessary correction.
Point 9
Line 378, it should be “Fig.S4A,B”.
Answer
Thank you for your comment. We introduced the necessary correction.
Round 2
Reviewer 2 Report
Comments and Suggestions for Authors
While almost issues have been addressed, I have only only two minor comments:
1. In Major concern #2 (at chapter 2.3.5), the authors should put citation(s) for that method.
2. In Major concern #12 (apoptosis & cell cycle), the authors should include their response into the Discussion.
Author Response
Comments and Suggestions for Authors
While almost issues have been addressed, I have only only two minor comments:
- In Major concern #2 (at chapter 2.3.5), the authors should put citation(s) for that method.
- In Major concern #12 (apoptosis & cell cycle), the authors should include their response into the Discussion.
Answer
Thank you very much. We appreciate the reviewers interest to our manuscript. In accordance to the marked points, we introduced the corrections in our manuscript.
- The citation [43] was added into 2.3.5 Material and Methods section.
- We introduced the paragraph about the cell cycle and apoptosis in discussion section. The changes were marked with color.